# Experimental and Computational Studies on the Interaction of DNA with Hesperetin Schiff Base Cu^II^ Complexes

**DOI:** 10.3390/ijms25105283

**Published:** 2024-05-13

**Authors:** Federico Pisanu, Anna Sykula, Giuseppe Sciortino, Feliu Maseras, Elzbieta Lodyga-Chruscinska, Eugenio Garribba

**Affiliations:** 1Dipartimento di Medicina, Chirurgia e Farmacia, Università di Sassari, Viale San Pietro, I-07100 Sassari, Italy; f.pisanu16@studenti.unica.it; 2Faculty of Biotechnology and Food Sciences, Institute of Natural Products and Cosmetics, Lodz University of Technology, Stefanowskiego 2/22, 90-537 Lodz, Poland; anna.sykula@p.lodz.pl (A.S.); elzbieta.lodyga-chruscinska@p.lodz.pl (E.L.-C.); 3Department de Química, Universitat Autònoma de Barcelona, Cerdanyola del Vallés, 08193 Barcelona, Spain; giuseppe.sciortino@uab.cat; 4Institute of Chemical Research of Catalonia (ICIQ), The Barcelona Institute of Science and Technology (BIST), 43007 Tarragona, Spain; fmaseras@iciq.es

**Keywords:** copper complexes, hesperetin Schiff bases, DNA interaction, computational calculations

## Abstract

The interactions with calf thymus DNA (CT-DNA) of three Schiff bases formed by the condensation of hesperetin with benzohydrazide (HHSB or L^1^H_3_), isoniazid (HIN or L^2^H_3_), or thiosemicarbazide (HTSC or L^3^H_3_) and their Cu^II^ complexes (CuHHSB, CuHIN, and CuHTSC with the general formula [CuL^n^H_2_(AcO)]) were evaluated in aqueous solution both experimentally and theoretically. UV–Vis studies indicate that the ligands and complexes exhibit hypochromism, which suggests helical ordering in the DNA helix. The intrinsic binding constants (*K*_b_) of the Cu compounds with CT-DNA, in the range (2.3–9.2) × 10^6^, from CuHTSC to CuHHSB, were higher than other copper-based potential drugs, suggesting that π–π stacking interaction due to the presence of the aromatic rings favors the binding. Thiazole orange (TO) assays confirmed that ligands and Cu complexes displace TO from the DNA binding site, quenching the fluorescence emission. DFT calculations allow for an assessment of the equilibrium between [Cu(L^n^H_2_)(AcO)] and [Cu(L^n^H_2_)(H_2_O)]^+^, the tautomer that binds Cu^II^, amido (am) and not imido (im), and the coordination mode of HTSC (O^−^, N, S), instead of (O^−^, N, NH_2_). The docking studies indicate that the intercalative is preferred over the minor groove binding to CT-DNA with the order [Cu(L^1^H_2_^am^)(AcO)] > [Cu(L^2^H_2_^am^)(AcO)] ≈ TO ≈ L^1^H_3_ > [Cu(L^3^H_2_^am^)(AcO)], in line with the experimental *K*_b_ constants, obtained from the UV–Vis spectroscopy. Moreover, dockings predict that the binding strength of [Cu(L^1^H_2_^am^)(AcO)] is larger than [Cu(L^1^H_2_^am^)(H_2_O)]^+^. Overall, the results suggest that when different enantiomers, tautomers, and donor sets are possible for a metal complex, a computational approach should be recommended to predict the type and strength of binding to DNA and, in general, to macromolecules.

## 1. Introduction

Transition metals have a significant impact on the functioning of living organisms due to their unique properties such as the interconversion of several oxidation states and coordination geometries and the electrochemical behavior. Moreover, they are active toward organic nucleophiles, enhancing the bioactivity of many organic ligands including Schiff bases [1,2,3,4,5,6,7,8]. 

Copper (Cu) is a transition metal that deserves to be mentioned. Indeed, it is an essential element that is involved in many biological processes such as heme synthesis, cellular respiration, redox and oxygenation reactions, and electron transfer [9,10,11,12,13]; at relatively low intracellular concentrations, copper acts as a key catalytic cofactor in a wide range of biological processes including mitochondrial respiration, antioxidant defense, and the synthesis of various biocompounds [14]. Moreover, the interest in the potential uses of copper in medicine has increased in the last twenty years, and several compounds have been tested, both in vitro and in vivo, as potential anticancer drugs [15,16,17,18,19,20,21]. The observation that tumor growth and metastasis require a higher demand for copper has given an extra boost to the research of Cu-related diagnostics and treatments in the fight against cancer [22,23,24,25]. Today, copper is one of the most widely used metals of the first transition series to develop anticancer drugs due to its redox nature, biocompatible properties, and high effectiveness in inducing cancer cell death [22,26]. For instance, three novel Cu^2+^ complexes bearing N,N,O-chelating salphen-like ligands affected HeLa cells to an extent similar to cisplatin and significantly better than carboplatin [27]. The compound Cu^II^–elesclomol, where elesclomol is *N*-malonyl-bis(*N*-methyl-*N*-thiobenzoyl hydrazide) in its doubly deprotonated form, has been proposed for metastatic melanoma [28], and the species Cu–triapine is active against many types of tumors [28,29]. In addition to their use as potential anticancer agents, the study of the properties of copper complexes has led to the development of non-steroidal anti-inflammatory drugs [18]. 

Various Cu compounds have reached clinical trials. The combination of copper or copper–gluconato with disulfiram (tetraethylthiuram disulfide) was proposed against various tumors and for newly diagnosed glioblastoma multiform (phase 2, identifier NCT03363659) [28,30], metastatic pancreatic cancer (phase 2, NCT03714555), and metastatic breast cancer (phase 2, NCT03323346). The species Cu–ATSM, with ATSM being diacetyl-bis(4-methyl-3-thiosemicarbazone), is not only under phase 2 of clinical trials for the treatment of rectal cancer (NCT03951337) [25], but has also progressed to phase 2/3 (NCT04082832) for its use against the neurodegenerative disease amyotrophic lateral sclerosis [31]. The complex ^64^Cu–DOTA, where DOTA is 2,2′,2″,2‴-(1,4,7,10-tetraazacyclododecane-1,4,7,10-tetrayl)tetraacetate, reached phase 1 (NCT02708511) for positron emission tomography-computed tomography use in imaging patients with ovarian and breast cancer [32]. The species Cu–histidinato is at phase 1/2 of the trials for its employment in the treatment of Menkes disease (NCT00001262). Finally, Casiopeinas^®^, formed by Cu^II^, 1,10-phenanthroline or 2,2′-bipyridine or one their derivatives, plus a monoanionic non-toxic ligand like glycinate or acetylacetonate, has been proposed for the treatment of breast and colon cancer [28,33,34,35]; the compound CasIII-ia, [Cu^II^(Me_2_bipy)(acac)]^+^, where Me_2_bipy is 4,4′-dimethyl-2,2′-bipyridine and acac is acetylacetonate, is currently undergoing phase 1 clinical trials in Mexico [36].

The exact mode of action of copper-based potential drugs is not always clear. They act with different mechanisms such as the inhibition of proteasome activity [37,38], telomerase activity [39], the formation of reactive oxygen species (ROS) [40,41], and, in particular, DNA interaction [42,43]. For this reason, the research on the interactions of copper complexes with DNA could be very important in biotechnology, pharmacology, and medicine for discovering and developing new potential drugs. 

For a metallocompound, binding to DNA can occur in a variety of ways. Overall, they can be classified into two general categories that comprise covalent and non-covalent binding [44]; in this last case, intercalation between base pairs and minor or major DNA groove-binding interactions are involved [45,46]. Copper complexes give mostly non-covalent interactions, either by intercalation, electrostatic attraction, and/or groove binding [47]. DNA intercalation has been reported in many studies [48,49,50,51], and this results in an inhibitory action on topoisomerases [20] and in a nuclease activity with the break of the double-strand of DNA [47,48]. In particular, the capability of copper complexes to behave as artificial nuclease can be utilized to design new potential anticancer therapeutics, and numerous cases of Cu complexes with outstanding anticancer activity and a lower toxicity profile than conventional Pt drugs have been reported in the literature [17,47,52]. Such behavior is shown, for example, by the family of Casiopeines^®^ [53]. For example, the aforementioned CasIII-ia and CasII-gly ([Cu^II^(Me_2_phen)(Gly)]^+^ with Me_2_phen = 4,7-dimethyl-1,10-phenanthroline and Gly = glycinate) exhibit higher activity and lower toxicity compared to cisplatin. CasIII-Ea, with formula [Cu^II^(Me_2_phen)(acac)]^+^, shows IC_50_ values of 4.9 and 2.1 μM on MCF-7 (breast cancer) and HCT-15 (colon cancer) cell lines, respectively, compared with 5.6 and 21.8 μM measured for cisplatin [54]; on the other hand, low toxicity is observed in non-tumor cells with accelerated growth like 3T3-L1 (healthy mice fibroblasts), treated with the CasIII-Ea compound [55]. As a final comment, it must be observed that binding between DNA and a metal complex can constitute hybrid catalysts in which transition metal complexes are embedded in a biomolecular scaffold represented by DNA; as a result of the formation of the new catalytic system, the reaction will proceed enantioselectively, and will optimally result in an additional acceleration [56]. 

Among the copper complexes, those formed by Schiff bases have been the object of extensive study. These compounds have attracted considerable interest in the scientific community due to their interesting properties, particularly those related to their biological and pharmacological action and possible use in medicine [1,2,3,4,5,6,7,8,57,58,59,60,61,62,63]. For example, hesperetin Schiff bases containing benzohydrazide (HHSB or L^1^H_3_), isoniazid (HIN or L^2^H_3_), and thiosemicarbazide (HTSC or L^3^H_3_) and their complexes with copper(II) have been investigated in analytical/spectral studies, and biological action in vitro such as cytotoxicity against human cancer cells, genotoxicity, and antimicrobial activity was ascertained by this and other groups [64,65,66]. Hesperetin (HESP) and its copper complex, CuHESP, were studied in some cases for comparison. The CuHHSB complex acts as a chemical nuclease during the cleavage of plasmid DNA in aqueous solution and is more effective than the free ligand HHSB against HeLa and K562 (human erythroleukemia) cells. In addition, among the tested bacterial strains, CuHHSB is very active toward *Staphylococcus aureus* [65]. All three copper complexes (CuHHSB, CuHIN, and CuHTSC) have an oxidative damaging effect greater than the ligands, with CuHIN showing the most prominent oxidative activity [66]. From our previous studies, it was found that modification of the structure of hesperetin increases its biological activity, for example, antioxidant activity, and for this reason, in the present work, we decided to perform another series of investigations to verify whether other properties of the modified ligands depended on the structural features. In this study, to allow for a better understanding of their potential biological effects, the interactions with DNA of the three hesperetin Schiff bases (HHSB, HTSC, and HIN) and their Cu^II^ compounds (CuHHSB, CuHIN, and CuHTSC) were screened in an aqueous solution both experimentally and theoretically. HESP and CuHESP were evaluated for comparison. UV–Vis and fluorescence spectra were carried out to obtain the binding characteristics; DFT methods were implemented to determine the structure of the most abundant species in solution, considering the exchange equilibrium between the monodentate equatorial ligand (AcO^−^) and solvent (H_2_O), the amido–imido tautomerism of the Schiff base ligands and the donor set (which may be (O^−^, N, S) or (O^−^, N, NH_2_) for L^3^H_3_); docking calculations allowed us to assess the binding to DNA and the effect of the nature of the substituents, unveiling new insights on the action of the copper complexes. The results could be useful for accelerating the development of this class of Cu-based drugs.

## 2. Results

### 2.1. Experimental Studies

#### 2.1.1. Behavior of Cu^II^ Complexes in Aqueous Solution

The rationale for the interaction of Schiff base ligands HHSB, HTSC, and HIN with Cu^2+^ ions was presented in a previous publication [66]. Spectral data (FTIR, UV–Vis, EPR, ESI-MS) and electrochemical techniques showed that, in the acetate complexes, the tested Schiff bases act as neutral tridentate ligand coordinating to Cu^2+^ through two oxygens or oxygen/sulfur plus a nitrogen donor atom. EPR measurements indicated that, in solution, the complexes keep their structures, with the ligands remaining bound to Cu^2+^ ions in a tridentate fashion with the (O^−^, N, O_ket_) or (O^−^, N, S) donor set.

The three tridentate Schiff base chelating ligands in the fully protonated form can be indicated as L^n^H_3_, with n = 1–3. These were obtained from the functionalization of the hesperetin with three different hydrazides, benzohydrazide (HHSB or L^1^H_3_) [65], isoniazid or pyridine-4-carbohydrazide (HIN or L^2^H_3_), and thiosemicarbazide (HTSC or L^3^H_3_) [67] (Figure 1). The three titratable protons in aqueous solution were those on the OH groups in positions 5 and 7 of the ring A and 3′ of the flavonoid moiety (ring B). 

With Cu^II^, they form different complexes in aqueous solutions, from [Cu(LH_2_)]^+^ to [Cu(L^n^H)], [Cu(L^n^)]^−^, and [Cu(L^n^H_–1_)]^2−^. When starting from Cu(AcO)_2_·H_2_O, one solid compound is formed, [Cu(L^n^H_2_)(AcO)], around pH 6–7. They are also indicated as CuHHSB, CuHIN, and CuHTSC (Figure 2).

Using EPR spectroscopy, it was demonstrated that the equilibrium [Cu(L^n^H_2_)(AcO)] + H_2_O/Solv ⇄ [Cu(L^n^H_2_)(H_2_O/Solv)]^+^ + AcO^−^ is established in water-containing mixtures or in an organic solvent (Solv) like DMSO or DMF. From [Cu(L^n^H_2_)(AcO)] to [Cu(L^n^H_2_)(H_2_O/Solv)]^+^, the value of *g*_z_ increases and *A*_z_(Cu) decreases by about (6–10) × 10^−4^ cm^−1^ [66]. The different steric hindrance of acetato and aqua ligand and the charge of the species could result in a different interaction with DNA. 

Moreover, the coordination of L^n^H_2_^−^ ligands can occur in the amido (am) or imido (im) form (Figure 3). In this case, the presence of a negative charge on the CO group and of NH instead of N atom in the five-membered chelate ring could also yield a different type of interaction with the double strand of DNA. 

Finally, both *R* and *S* enantiomers of the HHSB, HIN, and HTSC ligands, due to the stereogenic carbon in position 2 of the C ring of hesperetin (indicated with an asterisk in Figure 1), could interact with DNA with significant differences in the structural complementarity, and hence in terms of energy binding. 

#### 2.1.2. UV–Vis Studies on the DNA Binding

Absorption titrations were carried out to determine the DNA binding of the ligands and their Cu^II^ complexes in tris(hydroxymethyl)aminomethane (Tris) buffer. The UV–Vis absorption spectra of compounds in the absence and presence of calf thymus DNA (CT-DNA) are shown in Figure 4.

If the interaction of a metal species with DNA is by intercalation, the π*- and π-orbital of the base pairs may couple, resulting in a decrease of the π–π* transition energy and giving rise to a red shift (bathochromic effect) and a decrease in absorption (hypochromic effect). The hypochromism of the π–π* transition is often employed to find the binding constant between a metal species and CT-DNA, according to Equation (1).
(1)$$\frac{\left[\mathrm{CT}-\mathrm{DNA}\right]}{\varepsilon_{a}-\varepsilon_{f}}=\frac{\left[\mathrm{CT}-\mathrm{DNA}\right]}{\varepsilon_{b}-\varepsilon_{f}}+\frac{1}{K_{b}\left(\varepsilon_{b}-\varepsilon_{f}\right)}$$

In Equation (1), [CT-DNA] is the base pairs concentration, ε_a_ is the apparent extinction coefficient at a given concentration, while ε_f_ and ε_b_ are the coefficients of free and fully bound metal species, respectively [68,69]. 

The results obtained in this study indicate that the spectral profiles of the ligands and complexes are very different. After the addition of CT-DNA, the absorption bands of HHSB, HIN, HTSC and the corresponding copper complexes exhibited hypochromism, of which the highest one was for HHSB, but to a lower extent for the HIN, HTSC, and Cu species. Red shift was not observed. Hypochromic shift in the spectra of the compounds suggests helical ordering of both ligands and Cu compounds in the DNA helix. The absorption spectra of HHSB and HTSC when titrated with CT-DNA also showed an isosbestic point. The intrinsic binding constants (*K*_b_) of the compounds with CT-DNA are reported in Table 1. Notably, the values found for Cu^II^ complexes, in the range (2.3–9.2) × 10^6^, were higher than other copper-based compounds, for example, those formed by amidino-*O*-methylurea derivatives ((0.6–1.2) × 10^5^, ref. [70]), non-steroidal anti-inflammatory drugs piroxicam and lornoxicam ((2.7–3.4) × 10^4^, refs. [71,72]), Schiff base salicylaldehyde or 2-hydroxy-1-naphthalidene and L-valine and plus 1,10-phenanthroline ((5.7–6.5) × 10^3^, refs. [73,74]), and N-salicyl-*β*-amino alcohol Schiff bases ((0.04–2.4) × 10^6^, ref. [75]). The possibility of π–π stacking interaction due to the presence of aromatic rings and extended electronic delocalization favors the binding of CuHHSB, CuHIN, and CuHTSC, as confirmed by the high *K*_b_ values of the ligands ((3.7–6.9) × 10^6^, Table 1). Notably, the values determined for the ligands and Cu complexes were higher than the *K*_b_ of the adduct EB–DNA (1.2 × 10^5^, ref. [76]), where EB is ethidium bromide, a compound known for its capability to behave as an intercalative agent in the DNA double strand.

The results of the UV–Vis measurements and emission spectra presented in this study indicate the pro-oxidant activity of the tested compounds, which we previously demonstrated using the alkaline comet test to assess DNA breakage in HeLa cells exposed to various concentrations of HHSB, HTSC, HIN, CuHHSB, CuHIN, and CuHTSC [66]. Among the ligands, the most pronounced oxidative activity was revealed for HTSC, and, among the complexes, for CuHIN. Generally, Cu^II^ species have a greater oxidative damaging effect than the ligands. Moreover, both hesperetin azomethine derivatives and their copper complexes disclosed significantly lower impact on DNA damage compared to cisplatin [66]. 

The standard binding free energy *G* was estimated using the equation ∆*G*° = −RTln*K*_b_, where R is the universal gas constant (8.314 J·mol^−1^·K^−1^) and T is the temperature in Kelvins [77]. From this equation, it follows that the higher the binding constant *K*_b_ between CT-DNA and the ligand or Cu complexes, the more negative the standard free energy. Based on the data in Table 1, all processes were spontaneous under our experimental conditions with constant pressure and temperature. The reactions with CT-DNA that occurred most spontaneously were those observed for HHSB and CuHHSB.

For the complexes but not for the ligands, a linear correlation was found between the values of the *K*_b_ binding constants and the DNA double strain breaks (DNA-DSBs) (Figure 5). This phenomenon suggests a different mechanism of interaction of Cu complexes and ligands with DNA.

#### 2.1.3. Thiazole Orange (TO) Displacement Assay

Thiazole orange (TO) is a good intercalating dye, and the structure of the DNA–TO intercalation complex has been studied by many authors [78,79,80]. The fluorescence of TO increases after intercalating with DNA; hence, if a compound intercalates into the helix of DNA, it would compete with TO for the DNA intercalation sites, leading to a significant decrease in fluorescence intensity [81,82].

The emission spectra of the DNA-TO system decreased after the separate addition of each of the compounds studied (Figure 6). This result indicates that they are able to replace TO in the DNA−TO adduct, resulting in the dissociation of TO and in a decrease in the emission intensity. In other words, all of the studied compounds, HHSB, HIN, HTSC, and their corresponding Cu^II^ complexes, can compete with TO for the intercalation sites of DNA, suggesting that they may intercalate into the helix of DNA. For these measurements, HESP and CuHESP were studied for comparison.

An analysis of the emission spectra of the DNA-TO system in the presence of an increasing concentration of the studied ligands and copper compounds was carried out in order to estimate the quenching % upon their addition, their quenching constant *K*_SV_, and apparent binding constant *K*_app_ to DNA (Table 2). The data were analyzed using the Stern–Volmer equation, which puts in relationship the fluorescence with and without the quencher (*F* and *F*_0_), respectively, the concentration of the quencher [*Q*] (in this case, the hesperetin Schiff base ligands or the Cu complexes) and the Stern–Volmer quenching constant (*K*_SV_, in M^−1^): (*F*_0_/*F*) = 1 + *K*_SV_[*Q*]. The value of the apparent binding constant (*K*_app_) can be found with the equation *K*_app_ = *K*_TO_ × [TO]/[C_50%_], where *K*_TO_ is 3.0 × 10^6^ M^−1^, [TO] is the experimental concentration of TO, and [C_50%_] is the concentration of the quencher that produces a fluorescence decrease of 50%. 

All of tested compounds reduced the fluorescence intensity, indicating that they are able to compete with TO for the same binding sites, or that they interact with DNA at different sites, but close to TO. Among the ligands and complexes, HIN and CuHTSC possessed the most quenching ability. From the values of *K*_SV_, quenching, and *K*_app_ presented in Table 2, the order of increasing quenching and binding strength of the studied compounds is as follows: CuHTSC > CuHESP > CuHIN > CuHHSB > HIN > HHSB > HTSC > HESP. The order of the apparent binding constants suggests that Cu^2+^ ions have a distinct effect on quenching compared to the ligands themselves. This is probably related to the influence of the type of substituent inserted into the hesperetin moiety and to the interaction of the modified molecule with Cu^2+^. In fact, it must be observed that the electronic density of the HOMO in HESP is mainly localized over the rings A and B, while it is on ring A on the =N–NH group and the O atom of ring C in the remaining hesperetin derivatives [67]; therefore, the existence of electron-withdrawing substituents in HIN, HHSB, and HTSC enhances the electronic delocalization and conjugation, possibly influencing the process of quenching and binding of the ligands and metal species to DNA. 

The apparent binding constants increased in the order opposite to DNA oxidative damage or DNA double strain breaks (DSBs) from the comet assay [66]; these observations may indicate that the free ligands and Cu^II^ species interact with slightly different mechanisms of action depending on the chemical environment. The values of *K*_app_ (around 10^6^) point to a preferential interaction with DNA through intercalation, but other pathways (electrostatic and/or minor/major groove) could occur.

Finally, it should be noted that the values of the *K*_app_ constants of CuHIN and CuHHSB were very close to each other. This suggests that the isoniazid moiety affects in a negligible mode the binding strength of the Cu^II^ complexes. 

### 2.2. Computational Studies

#### 2.2.1. Geometry Optimization and Speciation Analysis of the Copper Complexes

Previously published spectroscopic results suggest that, in solution, [Cu(L^n^H_2_)(AcO)] has a square planar geometry around Cu^2+^ ion with donors (O^−^, N, O/S) and the fourth coordination position filled by the co-ligand (i.e., acetate AcO^−^) or a solvent molecule in [Cu(L^n^H_2_)(H_2_O)]^+^, with the protons on the OH groups in positions 7 and 3′ [66]. The DFT-characterized structures agree with the experimental outcomes, as shown in Figure 7.

According to what was experimentally observed in solution [65,66], the three complexes can undergo ligand exchange of the acetate co-ligand with a solvent molecule and, additionally, both the amido and imido tautomers of the three ligands can be in equilibrium, each of them potentially chelating the Cu^2+^ ion. Therefore, a plethora of species must be considered when questioning which interacts with DNA.

In this work, the behavior of the Cu^II^ species in solution, particularly co-ligand/solvent replacement and tautomerism, were studied, aiming at the determination of the active species in the systems with DNA. For this purpose, the complex with L^1^H_3_ was selected as a model of the whole set of ligands.

The replacement of the acetato ligand by a water molecule was modeled as shown in Equation (2), and the Cartesian coordinates for the DFT-optimized structure are reported in Appendix A.
[Cu(L^1^H_2_)(AcO)]_aq_ + (H_2_O)_8 aq_ ⇄ [Cu(L^1^H_2_)(H_2_O)]^+^_aq_ + [(H_2_O)_7_·AcO]^−^_aq_(2)

For the sake of accuracy, the two exchanging partners in this equilibrium (i.e., H_2_O and AcO^−^) were solvated using a mixed explicit/continuum method, according to Bryantsev et al. [83]. The calculated Δ*G*_aq_ was 2.0 kcal mol^−1^, suggesting that the species are in equilibrium in solution at room temperature. 

The study of the tautomerism of the Schiff bases (Figure 8) was assessed by considering the relative stability of the Cu^II^ complexes of the respective amido or imido tautomers (Equation (3)) and assuming the exchange of a proton with the solvent. 

The following reaction was studied: [Cu(L^1^H_2_^am^)(H_2_O)]^+^_aq_ + (H_2_O)_14 aq_ ⇄ [Cu(L^1^H^im^)(H_2_O)]_aq_ + [(H_2_O)_14_·H]^+^_aq_(3)

The Cartesian coordinates for the species involved in Equation (3), optimized by DFT methods, are reported in Appendix A. The calculated Δ*G*_aq_ of 7.8 kcal mol^−1^ allowed us to exclude the presence of the imido form of the Cu^II^ complex in aqueous solution. This is in line with the characterization in the solid state and with the experimental EPR results (Table 3 and ref. [66]).

The ligand L^3^H_3_ deserves a deeper consideration compared to the other ones. Indeed, its coordination mode is ambiguous, (O^−^, N, S) or (O^−^, N, NH_2_), as shown in Figure 9. The computed Δ*G*_aq_ for the two conformational isomers was 2.1 kcal·mol^−1^ with an energy barrier of 14.8 kcal·mol^−1^, ensuring their equilibrium at room temperature (Figure 9). 

The S-coordination is preferred over the N-coordination with a Δ*G*_aq_ for Equation (4) of 9.3 kcal mol^−1^, suggesting that the equilibrium between the two linkage isomers can be considered totally shifted toward the S-coordinated species. The Cartesian coordinates for [Cu(L^3^H_2_-κS)(AcO)]_aq_ and [Cu(L^3^H_2_-κN)(AcO)]_aq_ are listed in Appendix A.
[Cu(L^3^H_2_-κS)(AcO)]_aq_ ⇄ [Cu(L^3^H_2_-κN)(AcO)]_aq_
(4)

The computed spin Hamiltonian EPR parameters are in line with the experimental outcomes that suggested the S-coordination to be the favored in solution [66]. The predicted parameters are listed in Table 3.

Regarding the fourth coordination position, our DFT simulations confirmed the monodentate coordination of the acetato co-ligand with the non-coordinating oxygen perpendicular to the plane of the complex. The low energy barrier related to the flipping of the acetate, 2.9 kcal·mol^−1^, also ensures an equilibrium between two isoenergetic conformational isomers (Figure 10 and Appendix A). The coordinates for the two conformers are presented in Appendix A.

In general, the overall calculations allow us to discriminate which species are significantly present in solution. On the one hand, the amido–imido tautomerism of each complex and the linkage isomerism (–S or –NH_2_ donor) in the case of L^3^H_3_ were both significantly shifted toward only one of the two species (i.e., the amido tautomer and the –S donor, respectively). Thus, every imido tautomer as well as the coordination set (O^−^, N, NH_2_) for L^3^H_3_ can be ruled out from the docking calculations. On the other hand, the ∆*G* for the exchange of AcO^–^ with solvent indicates that the two species [Cu(L^n^H_2_^am^)(AcO)] and [Cu(L^n^H_2_^am^)(H_2_O)]^+^ have comparable concentrations in solution. Torsional freedom for the Cu–OAc and Cu–OH_2_ bonds was accounted for. 

#### 2.2.2. Docking with DNA

For hesperetin acting as a stereogenic unit in all the complexes due to its asymmetric C2 atom on the ring C (see Figure 1), docking calculations for the L^1^H_3_ ligand and complexes were run with both the *R* and *S* enantiomers as the benchmark. Table 4 summarizes the species docked with DNA. In Appendix A, the MOL2 files of each ligand structure implemented for the docking calculations are reported. In Appendix A, the MOL2 files of each ligand structure implemented for the docking calculations are listed (from Appendix A), while Appendix A contains the tables displaying the whole solutions and clusters for each calculation (from Appendix A).

For our systems, the interaction with DNA could occur in several modes: intercalative binding occurs through the insertion of a Cu complex, possibly positively charged, with aromatic ring(s) between two adjacent base pairs, stabilized by π–π stacking between the ring and the base pairs; groove binding with the reversible interaction of the copper species with a structure complementary to DNA major; and minor groove, which can vary with the size and shape of the groove [45].

We set up two different docking assays for the study of groove and intercalative binding modes. The structures of DNA crystallized with the pyrrolo[2,1-c][1,4]benzodiazepines derivative (PDB code 2K4L [84]) and intercalated by a dimeric derivative of the thiazole orange cation, TO (PDB code 108D [85]), were used as model receptors for the groove and intercalative binding assays, respectively. For the purpose of comparison, docking was additionally assessed for TO and the free ligand L^1^H_3_.

The results, gathered in Table 5, suggest that both the binding modes displayed high *Fitness* values, with intercalation favored for all species. The intercalative affinity order was [Cu(L^1^H_2_^am^)(AcO)] > [Cu(L^2^H_2_^am^)(AcO)] ≈ TO ≈ L^1^H_3_ > [Cu(L^3^H_2_^am^)(AcO)], in line with the experimental binding constants, *K*_b_, obtained from the UV–Vis spectroscopy (Table 1).

The main type of interactions found with the docking results for minor groove and intercalative binding modes were van der Waals contacts. These are more effective in the intercalative binding model because the DNA structure allows the ligand or metal species to enter the cavity between the base pairs (C_3_T_4_:G_14_A_13_ or A_5_G_6_:T_12_C_11_) to maximize these interactions, which can be described in terms of π–π contacts between the base pairs and the aromatic moieties of the hesperetin Schiff base ligands. In the groove binding model, ligands and metal moieties are not allowed to intercalate, and smaller *F* were observed. From the analysis of the four additive terms constituting the scoring function (see Equation (5) in Section 3.4), it is clear that the most important term among the four is *S*_vdw_ext_ (i.e., the one accounting for van der Waals interactions between the ligand and CT-DNA) (Table 5 and Appendix A). Conversely, the terms accounting for the hydrogen bonds, *S*_hb_ext_, and the intramolecular van der Waals forces plus the torsional strain energy, *S*_int_, provides a minor contribution or are not relevant. The term that considers the intramolecular hydrogen bonds, *S*_hb_int_, is zero in all cases. From a structural point of view, the extent of the aromaticity seems to play a major role in binding stabilization, with the more extended aromatic systems (i.e., L^1^H_3_ and L^2^H_3_) being favored to the smaller (i.e., L^3^H_3_). Moreover, the *R* enantiomers were slightly favored in all cases.

Comparing the two AcO^–^-and H_2_O-containing complexes, the respective order of affinity was [Cu(L^1^H_2_^am^)(AcO)] > [Cu(L^1^H_2_^am^)(H_2_O)]^+^, highlighting the relevance of the co-ligand in adduct stabilization.

The formation of DNA adducts can be rationalized as a multistep process, in which minor groove binding is the first step. From this intermediate, the adduct evolves toward the more stable intercalated adduct. This is described in Figure 11.

## 3. Materials and Methods

### 3.1. Synthesis

The synthesis and characterization of the CuHHSB complex have been presented in ref. [65], while the synthesis and description of the complexes CuHIN and CuHTSC were reported in ref. [66]. CuHESP was synthesized according to ref. [86]. All ligands (HHSB, HIN, and HTSC) were prepared according to the literature procedure [64]. The racemic hesperetin, benzohydrazide, 2-aminobenzohydrazide, isoniazid, thiosemicarbazide, and copper(II) acetate monohydrate (Cu(AcO)_2_·H_2_O) were purchased from Sigma-Aldrich Co. (Poznań, Poland). Thiazole orange and calf thymus DNA were purchased from Sigma-Aldrich Co. (St. Louis, MO, USA). All reagents were of analytical quality and used without further purification.

### 3.2. Spectroscopic Measurements

All absorption spectra were recorded at room temperature (25 °C) using matched quartz cells of 1.0 cm path length with a Varian UV–Visible Perkin-Elmer Lambda 11 spectrophotometer. Absorption spectra of 25 μM HHSB, HIN, HTSC, and complexes CuHHSB, CuHIN, CuHTSC, and the changes in their respective spectra on the subsequent addition of CT-DNA from 2.5 μM to 25 (35) μM in increasing concentration were recorded. All experiments were carried out in Tris buffer (5 mM Tris-HCl, 50 mM NaCl, pH 7.2). While measuring the absorption spectra, the solutions were allowed to incubate for 10 min before the absorption spectra were recorded, and an equal amount of CT-DNA was added to both the compound solution for the reference solution to eliminate the absorbance of CT-DNA itself. The intrinsic binding constant (*K*_b_) of our compounds with CT-DNA was obtained using Equation (1) [68,69,87,88]. Spectrofluorimetric measurements were performed on a Hitachi spectrophotometer, model F-2000. Tris buffer containing 25 µM DNA and 25 µM TO was titrated with solutions of the tested ligands or Cu^II^ complexes (concentrations in the range of 10–300 µM were used).

The inner filter correction was included in the values of fluorescence intensity in the emission spectra, as shown in Figure 6. The inner filter correction was calculated according to the equation $F_{obs}=F_{corr}\cdot 10^{\frac{A_{ex}\cdot d_{ex}}{2}\cdot \frac{A_{em}\cdot d_{em}}{2}}$, where *F_obs_* is the measured fluorescence, *F_corr_* is the correct fluorescence intensity that would be measured in the absence of inner-filter effects, *d_ex_* and *d_em_* are the cuvette pathlength in the excitation and emission direction (1 cm), respectively, and *A_ex_* and *A_em_* are the measured absorbance values at the excitation and emission wavelength, respectively, caused by ligand addition [89]. The emission spectra were recorded in the 500–600 nm (emission) wavelength range at an excitation wavelength λ_ex_ of 430 nm. Measurements were performed in a quartz cuvette at room temperature.

### 3.3. DFT Calculations

Geometry optimization and vibrational frequency calculations were run by Gaussian 16 [90] using the DFT method with the B3LYP functional and Grimme’s D3 correction [91]. The basis-set 6-31G(d,p) was applied to the main group elements, while the SDD plus f-functions was employed for the metal [92]. The SMD continuum model for water was used to account for the aqueous environment [93]. Frequency calculations allowed us to obtain the thermal and entropic corrections. To determine the Gibbs free energies, these corrections were added to the potential energy obtained by single-point calculations on the optimized structures, with the basis-set def2-TZVP for the main group and def2-QZVP for the d-block elements [94]. A mixed explicit/continuum model [83] was used for aqueous water molecules, hydronium, and acetate. A dataset collection of the computational results is available in the ioChem-BD repository and can be accessed via https://doi.org/10.19061/iochem-bd-1-322 [95].

The EPR parameters were computed using the ORCA program [96,97,98]. The combination of the PBE0 [99] and 6-311G(d,p) basis-set was used to calculate the ***g*** tensor, while the B3LYP [100,101] was applied to determine the ***A*** tensor with the same basis-set, following the published results of Sciortino et al. [102].

### 3.4. Docking Calculations

Docking calculations were performed by GOLD 5.3 [103]. Previously, the original ligands of the DNA crystallographic structures (PDB code 2K4L [84] for the minor groove binding model; PDB code 108D [85] for the intercalation model) were removed and hydrogen atoms were added using UCSF Chimera [104].

The fluorescent dye thiazole orange (TO), used in the fluorescent quenching experiments, was optimized with the aforementioned method and used as the docking ligand for the purpose of comparison. The binding site comprised the whole structure for both models (PDB codes 2K4L [84] and 108D [85], respectively). For each ligand, 100 genetic algorithm (GA) runs were processed. Free rotation of the AcO–Cu coordination bond was considered along the simulations.

GoldScore, the scoring function used to calculate the *Fitness* score *(F*) in this study, has already been validated for docking with metal complexes as ligands (it must be noted that, in docking terminology, the species interacting with the receptor is generically named as the ligand) [105,106]. The polynomial scoring function consists of a sum of four terms (Equation (5)), accounting for hydrogen bonds between the ligand and receptor (*S*_hb_ext_), van del Waals interactions between the ligand and receptor (*S*_vdw_ext_), intramolecular hydrogen bonds of the ligand (*S*_hb_int_), and a term that summarizes the intramolecular van der Waals forces of the ligand and its torsional strain energy (*S*_int_). Each term is weighted by the coefficients α = 1, β = 1.375, γ = 1, and δ =1.
Fitness score *(F) =* α *S*_hb_ext_ + β *S*_vdw_ext_ + γ *S*_hb_int_ + δ *S*_int_
(5)

The GA was set automatically with a minimum 1 × 10^5^ operations. All other settings were left as the default. Finally, the docking solution were clustered by GOLD using an RMSD with a threshold of 2.5 Å.

## 4. Conclusions

Among the antitumor drugs, after the discovery of cisplatin and the development of its derivatives, metal complexes have gained a broad space in experimentation, and Cu-based potential therapeutics are worth being mentioned for their high activity and low toxicity. The binding studies of metal complexes with DNA are at the basis of the development of existing or new metal-based drugs. For metal complexes formed by first-row transition elements, these studies present several limitations: the possibility of exchange reactions of the ligands with water or solvent molecules, the formation of two or more isomers and of the corresponding enantiomers, the variation in the coordination number, and the possibility of the existence of tautomeric forms for the organic ligands can result in a wide variety of species able to interact with DNA. Therefore, it is not trivial to interpret the experimental results of the interaction studies, often based on UV–Vis or fluorescence spectroscopy. Some authors have noticed that the scarce knowledge of these aspects could be at the basis of the lack of interest by pharmaceutical companies for metal-based drugs compared with organic compounds, and the failed development of many of them [107]. Therefore, all approaches useful to characterize the metal–protein binding, preferably based on multiple techniques and a combined application of experimental and computational techniques, are desirable.

The systems discussed in the present study, containing a Schiff base formed by hesperetin with benzohydrazide (HHSB), isoniazid (HIN), and thiosemicarbazide (HTSC) represent a good example of what was above-mentioned. In fact, the solid compounds with the general formula [Cu(L^n^H_2_)(AcO)] can exchange the acetate ion with water, the ligands can bind the Cu centers in the amido or imido form, and HTSC can coordinate with the (O^−^, N, NH_2_) or (O^−^, N, S) donor set. The combined approach of the DFT and docking methods allowed us to demonstrate that: (i) an equilibrium between [Cu(L^n^H_2_)(AcO)] and [Cu(L^n^H_2_)(H_2_O)]^+^ exists in aqueous solution, and so both species can interact with DNA; (ii) the amido and not imido tautomer of HHSB, HIN, and HTSC binds Cu^2+^ ion; and (iii) the coordination mode of HTSC is (O^−^, N, S) and not (O^−^, N, NH_2_). Moreover, the results indicate that the intercalative binding is stronger than minor groove interaction with the order [Cu(L^1^H_2_^am^)(AcO)] > [Cu(L^2^H_2_^am^)(AcO)] ≈ TO ≈ L^1^H_3_ > [Cu(L^3^H_2_^am^)(AcO)], in agreement with the experimental binding constants to DNA (*K*_b_) obtained from UV–Vis spectroscopy, with the aromaticity of the ligands playing a major role in binding stabilization. The computational data also suggest that the binding of [Cu(L^n^H_2_^am^)(AcO)] is preferred over [Cu(L^n^H_2_^am^)(H_2_O)]^+^, and the binding of *R* is favored compared to *S* enantiomers.

As a general conclusion, it must be highlighted that when different donor sets for the ligands, isomers, enantiomers, and tautomers for the metal complexes are possible, a computational approach should be recommended to predict the type and strength of binding to DNA, and in general, to macromolecules. To this it must be added, however, that a complete characterization of these systems is often not possible because a mixture of metal moieties can form and interact with DNA as well as because at the physiological metal concentration, not greater than a few μM, hydrolysis can lead to hydroxide species that could dominate in the solution at pH around 7.

## Figures and Tables

**Figure 1 ijms-25-05283-f001:**
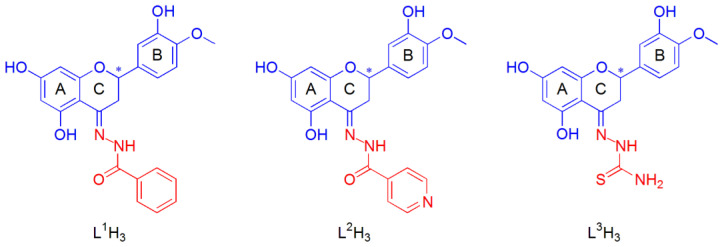
Structural formula of the free hesperetin Schiff base ligands: benzohydrazide (L^1^H_3_); isoniazid (L^2^H_3_); thiosemicarbazide (L^3^H_3_). A different color is used for the molecular moiety deriving from hesperetin (blue) and hydrazides (red). The asterisk indicates the stereogenic carbon atom in position 2 of the C ring of hesperetin.

**Figure 2 ijms-25-05283-f002:**
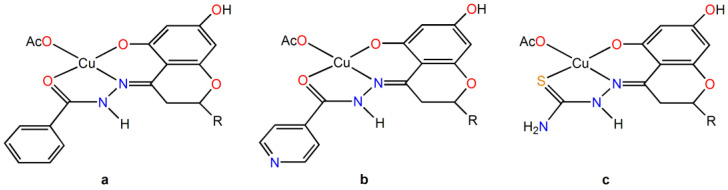
Structural formula of the complexes: (**a**) CuHHSB, (**b**) CuHIN, and (**c**) CuHTSC. The R group is the ring B of hesperetin with OH and OCH_3_ substituents on the 3′ and 4′ positions.

**Figure 3 ijms-25-05283-f003:**
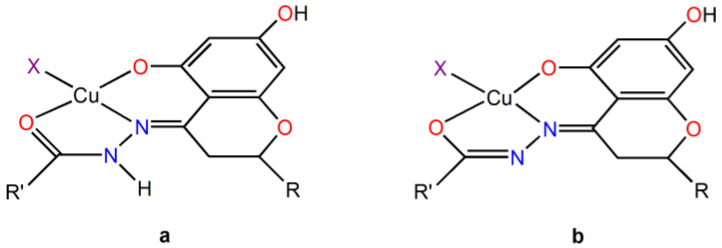
Structure for the amido (**a**) and imido (**b**) coordination for the CuHHSB, CuHIN, and CuHTSC complexes. X represents a water or a AcO^–^ ligand. The R group is the ring B of hesperetin with OH and OCH_3_ substituents on the 3′ and 4′ positions, while R’ stands for benzohydrazide, isoniazid, or thiosemicarbazide moieties of the HHSB, HIN, and HTSC ligands.

**Figure 4 ijms-25-05283-f004:**
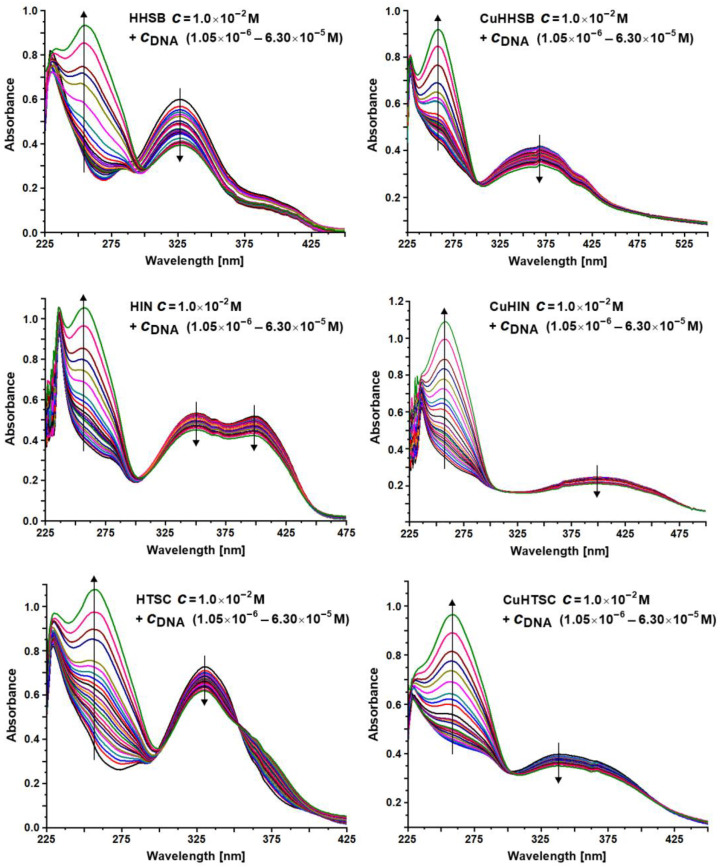
UV–Vis absorption spectra of the ligands and Cu^II^ compounds in the absence and presence of calf thymus DNA. The arrows show the changes in the absorbance after the addition of increasing amounts of CT-DNA.

**Figure 5 ijms-25-05283-f005:**
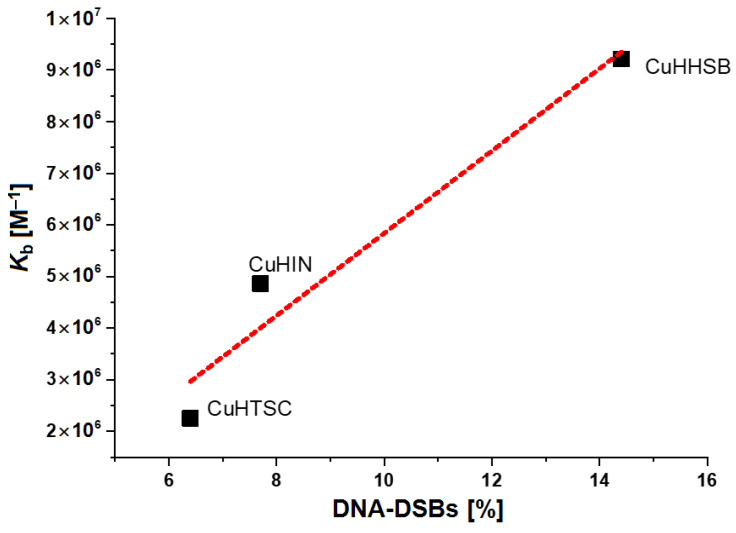
Relationship between the values of *K*_b_ and DNA-DSBs (DNA double-strain breaks) for Cu^II^ complexes. Data taken from ref. [66]. The linear equation to fit the experimental points is *y* = a + b*x*, with a = (−2.14 ± 1.88) × 10^6^ and b = (7.98 ± 1.85) × 10^5^. The value of R^2^ is 0.90.

**Figure 6 ijms-25-05283-f006:**
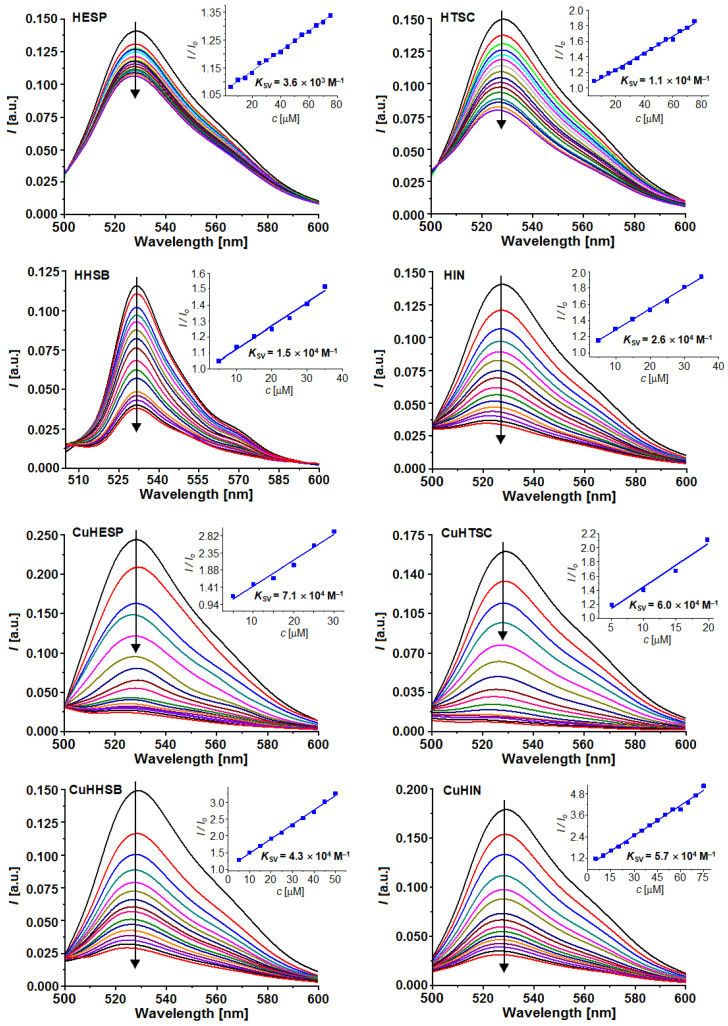
Fluorescence spectra of the DNA–TO adduct upon the addition of HESP, HHSB, HIN, HTSC, and their corresponding complexes, CuHESP, CuHHSB, CuHIN, and CuHTSC. The concentration of CT-DNA was 2.5 × 10^−5^ M, and that of TO was 1.0 × 10^−5^ M, while the concentrations of the ligands and complexes were in the range (0.5–7.5) × 10^−5^ M. Insets represent Stern–Volmer relationships. The arrows show the changes in the fluorescence after the addition of increasing amounts of CT-DNA.

**Figure 7 ijms-25-05283-f007:**
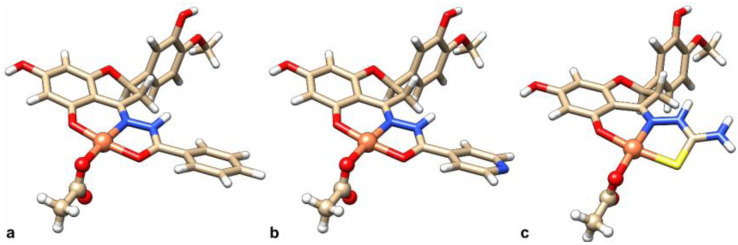
DFT-optimized geometry structures of the complexes: (**a**) [Cu(L^1^H_2_)(AcO)], (**b**) [Cu(L^2^H_2_)(AcO)], and (**c**) [Cu(L^3^H_2_)(AcO)].

**Figure 8 ijms-25-05283-f008:**
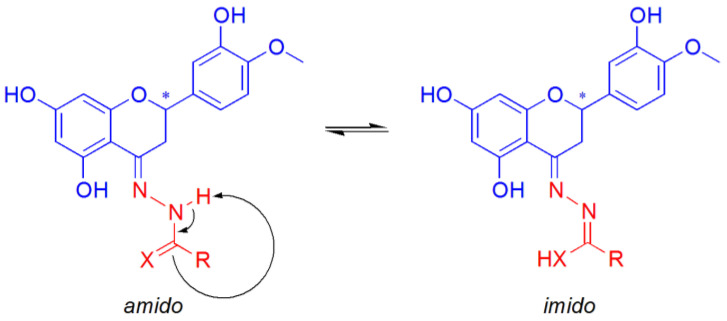
Schiff base ligand tautomerism. X represents an atom of O or S. The asterisk indicates the stereogenic carbon atom in position 2 of the C ring of hesperetin.

**Figure 9 ijms-25-05283-f009:**
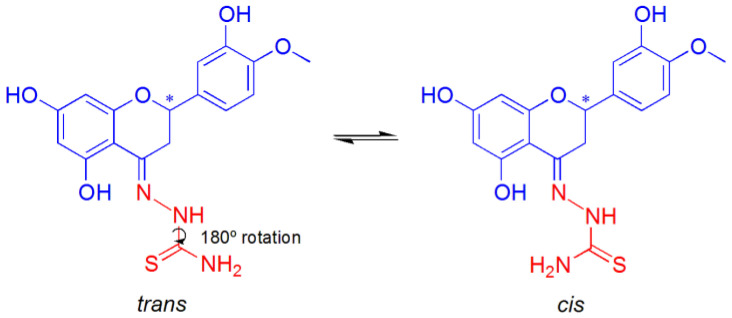
Isomerization of the ligand L^3^H_3_. The asterisk indicates the stereogenic carbon atom in position 2 of the C ring of hesperetin.

**Figure 10 ijms-25-05283-f010:**
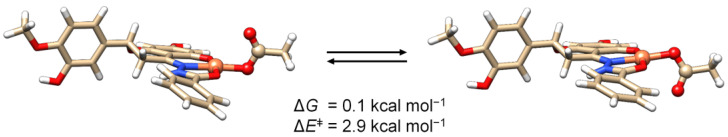
Interconversion between the most stable conformers of [Cu(L^3^H_2_^am^)(AcO)] and the relative energies. The values of Δ*G* and of the energy barrier Δ*E*^‡^ for the flipping of the acetato ligand are also shown.

**Figure 11 ijms-25-05283-f011:**
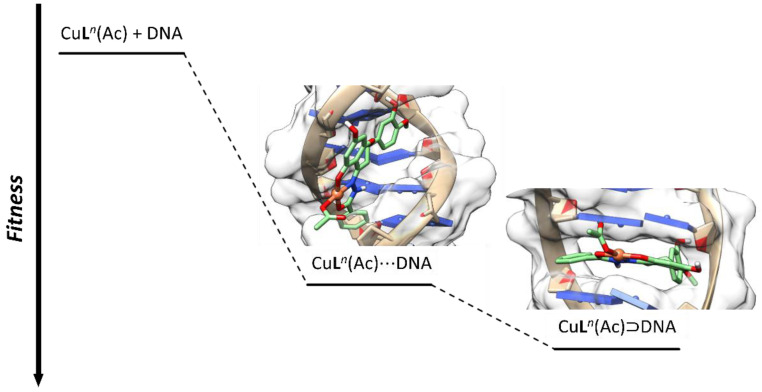
Schematized multistep binding process between DNA and [Cu(L^n^H_2_^am^)(AcO)], along with the *Fitness* values of the docking assays of the formed adducts. The symbols “⋯” and “⊃“ indicate, respectively, the minor groove and intercalative binding.

**Table 1 ijms-25-05283-t001:** Experimental data on the interaction of the HTSC, HHSB, and HIN ligands and their Cu^II^ complexes with DNA.

Compound	*K*_b_ ^1^	R^2^	∆*G*° ^2^	DSBs ^3^	Oxidative DNA Damage (Endo III) ^4^	*K*_app_ ^1^
HTSC	4.23 × 10^6^	0.994	−37.8	13.5 ± 0.7	10.2 ± 2.0	3.30 × 10^5^
HHSB	6.88 × 10^6^	0.980	−39.0	23.3 ± 1.0	7.5 ± 3.9	4.30 × 10^5^
HIN	3.70 × 10^6^	0.991	−37.5	19.5 ± 0.7	5.3 ± 3.5	7.94 × 10^5^
CuHTSC	2.25 × 10^6^	0.940	−36.2	6.4 ± 2.3	11.6 ± 1.9	1.60 × 10^6^
CuHHSB	9.21 × 10^6^	0.989	−39.7	14.4 ± 0.7	8.3 ± 2.7	1.36 × 10^6^
CuHIN	4.86 × 10^6^	0.992	−38.1	7.7 ± 3.6	12.2 ± 4.2	1.38 × 10^6^
EB ^5^	1.23 × 10^5^	^6^	−34.7	^6^	^6^	^6^

^1^ Values in M^–1^. ^2^ Values in kcal mol^–1^. ^3^ DNA double strand breaks (DSBs), taken from ref. [66]. ^4^ Oxidative DNA damage evaluated with the DNA repair enzyme–endonuclease III (Endo III), taken from ref. [66]. ^5^ From ref. [76]. ^6^ No data available.

**Table 2 ijms-25-05283-t002:** Thiazole orange (TO) quenching assay results for hesperetin Schiff base ligands and Cu^II^ complexes.

Compound	*K*_SV_ ^1^	R^2^	Quenching ^2^	C_50%_	*K*_app_ ^1^
HESP	(3.63 ± 0.04) × 10^3^	0.996	83	2.56 × 10^–4^	1.17 × 10^5^
HTSC	(1.08 ± 0.07) × 10^4^	0.994	69	9.10 × 10^–5^	3.30 × 10^5^
HHSB	(1.46 ± 0.05) × 10^4^	0.983	59	6.97 × 10^–5^	4.30 × 10^5^
HIN	(2.59 ± 0.07) × 10^4^	0.996	46	3.78 × 10^–5^	7.94 × 10^5^
CuHESP	(7.11 ± 0.05) × 10^4^	0.970	9	1.97 × 10^–5^	1.51 × 10^6^
CuHTSC	(6.01 ± 0.03) × 10^4^	0.960	5	1.87 × 10^–5^	1.60 × 10^6^
CuHHSB	(4.31 ± 0.08) × 10^4^	0.996	20	2.21 × 10^–5^	1.36 × 10^6^
CuHIN	(5.65 ± 0.03) × 10^4^	0.992	20	2.18 × 10^–5^	1.38 × 10^6^

^1^ Values in M^–1^. ^2^ %*F*_0_.

**Table 3 ijms-25-05283-t003:** Calculated and experimental EPR parameters, *g*_z_ and *A*_z_, of the species.

Compound	*g*_z_^calcd^ (*g*_z_^exptl^) ^1^	*A*_z_^calcd^ (*A*_z_^exptl^) ^1,2^	Error/% ^3^
[Cu(L^3^H_2_^am^)(AcO)]_aq_	2.204 (2.243)	187.9 (188.8)	–1.7; –0.5
[Cu(L^3^H_2_^am^)(H_2_O)]^+^_aq_	2.183 (2.243)	195.6 (188.8)	–2.7; +3.6
[Cu(L^3^H_2_^am^)(H_2_O)]^+^_aq_	2.165 (2.243)	201.7 (188.8)	–3.5; +6.9
[Cu(L^3^H_2_-κS^am^)(AcO)]_aq_	2.191 (2.200)	189.7 (185.0)	–0.4; +2.5
[Cu(L^3^H_2_-κS^am^)(H_2_O)]^+^_aq_	2.176 (2.200)	169.2 (185.0)	–1.1; +8.6

^1^ Calculated and experimental values (in parenthesis). ^2^ Values reported in 10^–4^ cm^–1^. ^3^ Relative error of *g*_z_ (on the left) and of *A*_z_ (on the right) calculated with the formula: [(Calcd. − Exptl.)/Exptl.] × 100.

**Table 4 ijms-25-05283-t004:** Species selected as ligands for the docking calculations.

Ligand	AcO^−^/H_2_O Exchange	Chirality
L^1^H_3_	L^1^H_3_^am^	(*R*)-L^1^H_3_^am^
		(*S*)-L^1^H_3_^am^
	[Cu(L^1^H_2_^am^)(AcO)]	[Cu((*R*)-L^1^H_2_^am^)(AcO)]
		[Cu((*S*)-L^1^H_2_^am^)(AcO)]
	[Cu(L^1^H_2_^am^)(H_2_O)]^+^	[Cu((*R*)-L^1^H_2_^am^)(H_2_O)]^+^
		[Cu((*S*)-L^1^H_2_^am^)(H_2_O)]^+^
L^2^H_3_	[Cu(L^2^H_2_^am^)(AcO)]	[Cu((*R*)-L^2^H_2_^am^)(AcO)]
	[Cu(L^2^H_2_^am^)(H_2_O)]^+^	[Cu((*R*)-L^2^H_2_^am^)(H_2_O)]^+^
L^3^H_3_	[Cu(L^3^H_2_-κS^am^)(AcO)]	[Cu((*R*)-L^3^H_2_-κS^am^)(AcO)]
	[Cu(L^3^H_2_-κS^am^)(H_2_O)]^+^	[Cu((*R*)-L^3^H_2_-κS^am^)(H_2_O)]^+^
TO	TO	TO

**Table 5 ijms-25-05283-t005:** Best GoldScore solutions for all of the Cu^II^ complexes and ligands with DNA for minor groove and intercalative binding. *Fitness* scores are sorted in order of decreasing values.

Interaction Mode	Ligand	*F*_max_ ^1^	*S*_hb_ext_ ^2^	*S*_vdw_ext_ ^3^	S_int_ ^4^	Pop. ^5^
Intercalation	[Cu((*R*)-L^2^H_2_^am^)(AcO)]	92.7	1.3	92.0	−0.6	85
Intercalation	[Cu((*R*)-L^1^H_2_^am^)(AcO)]	92.4	1.7	91.0	−0.3	83
Intercalation	[Cu((*R*)-L^1^H_2_^am^)(H_2_O)]^+^	89.5	0.0	89.7	−0.2	94
Intercalation	[Cu((*R*)-L^2^H_2_^am^)(H_2_O)]^+^	89.2	0.0	89.5	−0.3	89
Intercalation	[Cu((*S*)-L^1^H_2_^am^)(AcO)]	88.2	4.6	86.3	−2.7	10
Intercalation	TO	88.1	0.0	90.9	−2.8	5
Intercalation	(*R*)-L^1^H_3_	88.0	0.0	88.3	−0.3	51
Intercalation	(*S*)-L^1^H_3_	87.0	1.7	89.5	−4.3	40
Intercalation	[Cu((*S*)-L^1^H_2_^am^)(H_2_O)]^+^	84.5	0.0	87.7	−3.1	12
Intercalation	[Cu((*R*)-L^3^H_2_^am^)(AcO)]	83.2	0.4	83.4	−0.6	2
Minor groove binding	[Cu((*S*)-L^1^H_2_^am^)(AcO)]	75.9	6.6	71.7	−2.4	88
Minor groove binding	[Cu((*S*)-L^1^H_2_^am^)(H_2_O)]^+^	75.1	2.7	74.5	−2.2	2
Intercalation	[Cu((*R*)-L^3^H_2_^am^)(H_2_O)]^+^	75.5	2.0	80.0	−6.4	25
Minor groove binding	[Cu((*R*)-L^3^H_2_^am^)(AcO)]	74.5	1.8	74.7	−1.9	66
Minor groove binding	[Cu((*R*)-L^1^H_2_^am^)(AcO)]	72.8	7.1	73.4	−7.7	83
Minor groove binding	[Cu((*R*)-L^2^H_2_^am^)(AcO)]	70.3	0.1	72.0	−1.8	11
Minor groove binding	[Cu((*R*)-L^1^H_2_^am^)(H_2_O)]^+^	70.2	0.2	72.3	−2.2	42
Minor groove binding	[Cu((*R*)-L^2^H_2_^am^)(H_2_O)]^+^	69.0	3.5	67.7	−2.2	25
Minor groove binding	[Cu((*R*)-L^3^H_2_^am^)(H_2_O)]^+^	68.6	8.8	62.9	−3.0	9

^1^ Highest *Fitness* score. ^2^ Term accounting for hydrogen bonds between the ligand (in this study the hesperetin Schiff base or metal moiety) and receptor (in this study the CT-DNA). ^3^ Term accounting for van del Waals interactions between the ligand and receptor. ^4^ Term accounting for the intramolecular van der Waals forces of the ligand and its torsional strain energy. ^5^ Cluster population: number of solutions of the cluster with the highest *Fitness* scores. Clustering was performed using a root-mean-square-deviation (RMSD) with a threshold of 2.5 Å.

## Data Availability

Data is contained within the article and Appendix A.

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
