# Peer review of "Experimental and Computational Studies on the Interaction of DNA with Hesperetin Schiff Base CuII Complexes"

_ijms, 2024, doi:10.3390/ijms25105283_

Round 1

Reviewer 1 Report

Comments and Suggestions for Authors  

Reviewer’s Comments to the Authors 

The current research manuscript describes the experimental and computational studies on the interaction of  DNA with hesperetin Schiff base CuII complexes. In the current version, the interactions with DNA of the three hesperetin Schiff base CuII compounds  (CuHHSB,  CuHIN, and  CuHTSC) were evaluated in an aqueous solution both experimentally and theoretically in a concise manner. In the reviewer’s opinion, the manuscript has been appropriately organized. The reviewer feels that a potential reader can easily understand the details by consulting the main data outlined in the current version of the manuscript, with few exceptions. The write-up is clear and to the point which is crucial for a research article. However, in the current version, the reviewer feels that this work can advance further consideration after addressing the following comments.

Major Modifications

1)     In the title itself, …the interaction of DNA with hesperetin Schiff bases Cu II complexes, should be like this…. the interaction of DNA with hesperetin Schiff base Cu II complexes. ( note the word base)

2)     The title of the paper depicts experimental and computational studies on the interaction of DNA with hesperetin Schiff base CuII complexes. It is suggested that, in the introduction of this manuscript, the authors should only add those findings that are relevant to DNA binding and demonstrate its facts. From this perspective, it is advised to double-check references 1-7 that seem to be general and add them, as suggested above.

3)     In the introduction section, the authors have stated that interest in the potential use of copper in medicine has recently increased, and several compounds have been tested,  both in vitro and in vivo,  as potential anticancer drugs. Interestingly, the cited references were more than a decade old. Enormous technological advancements have been achieved since 2010. These old references should be replaced. Additionally, if authors are suggesting the potential utility of Cu complexes in medicine then it is recommended to add in the introduction part some Cu-containing drugs or potential molecules or druggable molecules that are moving in clinical trials or at various phases of FDA approval. Only a few basic invivo or invitro citations are not enough.

4)Table 2. shows thiazole orange (TO) quenching assay results for hesperitin  Schiff base ligands and Cu II complexes. Double-check the order in Table 2. Why Cu HESP is not listed in the order?

5) Correct the line 185.

6) Figure 8 demonstrates Schiff base ligand tautomerism. Please denote X.

7) The six-membered intramolecular hydrogen bonding is available in the structural formula of the free hesperetin and Schiff base ligands (L1H3, L2H3, and L3H3). What was the rationale for the interaction with Cu in the Cu complexes? This should be elaborated upon further.

8) It is described that the values found for Cu II complexes (2.3× 106 - 9.2) are higher than those of other Cu-based reported compounds (see Ref 70-75). This logic is associated with π-stacking interactions due to the aromatic rings favoring the binding of Cu Complexes. The reviewer believes that the Cu complexes in Ref 70-75 also have pi-stacking interactions due to their aromatic rings in them. How do the authors justify their rationale in this study?

9)How did the authors arrive at the concentration (25 µM)  used in the study? Did these complexes undergo any thorough systemic assessments at other concentrations before?  If yes, kindly provide a reference and describe it in the manuscript.

10) Was it necessary to perform emission spectra in a 2 mL cuvette? In addition, provide a reference for the wavelengths used ( 430 and 530 nm).

11) The authors stated that, among the potential antitumor drugs, after the discovery of cisplatin and the development of its derivatives, metal complexes gained a broad space in experimentation, and Cu-based potential therapeutics are worth mentioning for their high activity and low toxicity, which can bypass undesirable side effects and resistance phenomena due to the prolonged administration of other drugs.  To support this, authors should have added related Cu molecules to the manuscript.

12) In my view, the results and discussion sections are discussed in detail. However, the study outcomes have not been compared to those of similar studies already done. It is suggested to describe and compare previous similar/identical studies and then draw the conclusions of this study. What additional scientific advancements have been made and contributed specifically to this report? The discussion is usually about the comparison and proving that your research is more reliable than previous ones.

13) The conclusion again repeats the same lines already discussed in the abstract, results, and introduction. The conclusion should consist of your objectives being fulfilled, prospects, and limitations for researchers to take ideology from it. Please modify this accordingly and add possible limitations to future research.

Reviewer 2 Report

Comments and Suggestions for Authors

The manuscript, titled "Experimental and Computational Studies on the Interaction of DNA with Hesperetin Schiff Bases Cu(II) Complexes," describe the investigation of copper(II) complexes with three tridentate Schiff base chelating ligands derived from hesperetin. The study comprehensively explores various aspects, including ligand coordination, DNA binding properties, computational analysis of complex speciation, and interaction modes with DNA. While the manuscript is well-written, there are several points of consideration that need addressing before its acceptance for publication.

Firstly, the introduction section should be expanded to provide a clearer context for the research. This includes elucidating the significance of studying copper(II) complexes, their potential applications, and the rationale behind choosing hesperetin-derived ligands.

In the results section (2.1.1. Behavior of Cu(II) Complexes in Aqueous Solution), it is noted that this part merely describes findings from a prior publication by the same group. Therefore, it would be preferable to summarize these findings and integrate them into the discussion section, rather than presenting them as experimental work within this manuscript.

Furthermore, in Figure 1, the asterisk (*) sign should be explained in the caption to enhance clarity.

The section on UV-Vis studies on DNA binding (2.1.2) appears to be somewhat confusing. It would be beneficial to present the spectra of the free ligands and the Cu complexes separately. Additionally, reorganizing the results and breaking down the discussion into subsections corresponding to different aspects of the study, such as ligand coordination, DNA binding, and computational analysis, could significantly improve the clarity of the findings.

Lastly, in the conclusion section, the first two paragraphs seem more like an introductory part and should be removed. Instead, the authors should expand the conclusions based on the findings of this work, providing a concise summary of the key insights garnered from the study.

Comments on the Quality of English Language

The quality of English is ok. Some minor spelling mistakes scattered in the text should be addressed. 

Reviewer 3 Report

Comments and Suggestions for Authors

The manuscript addresses an interesting scientific topic and is well-written. However, I have some concerns regarding the control of the results obtained for both fluorescence and molecular docking. In my view, the manuscript could be accepted pending the inclusion of the inner filter correction for fluorescence in the article, along with the inclusion of the results and criteria chosen for molecular docking conformation.

-Figure 4 exhibits poor image quality. The ligand titration images are too small, hindering the reader's ability to analyse the spectra. The authors should enhance the quality and size of the figure.

-In the "Spectroscopic measurements" section, the authors omitted whether inner filter correction was applied in the fluorescence experiments to ensure that the fluorescence signal suppression effect is not merely an artifact. Did the authors conduct this correction? This information should be added to the manuscript.

-In the fluorescence experiments, the authors could determine the interaction free energy based on the affinity constant obtained, thereby indicating whether complexation is spontaneous. It would be valuable for the authors to include this information regarding the spontaneity of the complexation process.

-For the molecular docking simulations, cluster data and interaction energy scoring functions were not provided, making it unclear how conformations were selected from the calculations. The authors should display the docking graphs and specify the criteria for choosing conformations in the manuscript.

-Furthermore, concerning the results of molecular docking, the authors should identify the types of molecular interactions occurring with the DNA and discuss their significance in the manuscript.

-Lastly, the authors did not compare their results with those reported in the literature, which should be addressed in the manuscript.

Comments on the Quality of English Language

Minor editing of English language required

Round 2

Reviewer 1 Report

Comments and Suggestions for Authors

PAST COMMENT: 4) Table 2. shows thiazole orange (TO) quenching assay results for hesperitin Schiff base ligands and CuII complexes. Double-check the order in Table 2. Why Cu HESP is not listed in the order?

ANSWER. In Table 2 the order is now preserved. HESP is the first ligand, followed by HTSC, HHSB, HIN, CuHESP is the first complex followed by the other complexes, CuHTSC, CuHHSB, and CuHIN. We would like to add that the footnote 3 was removed because it was unrelated to the values presented in the table. We apologize for the error.

CURRENT COMMENT:  On one side author says that HESP and CuHESP were evaluated for comparison (Line 136). TABLE 2 also shows the obtained data but in lines ( 295 – 298), in the order of increasing quenching and binding strength of the studied compounds, still HESP is not included. This should be addressed and logical reason should be given.
